# The Spatiotemporal Distribution, Abundance, and Seasonal Dynamics of Cotton-Infesting Aphids in the Southern U.S.

**DOI:** 10.3390/insects14070639

**Published:** 2023-07-15

**Authors:** John W. Mahas, Jessica B. Mahas, Charles Ray, Adam Kesheimer, Todd D. Steury, Sophia R. Conzemius, Whitney Crow, Jeffrey Gore, Jeremy K. Greene, George G. Kennedy, David Kerns, Sean Malone, Silvana Paula-Moraes, Phillip Roberts, Scott D. Stewart, Sally Taylor, Michael Toews, Alana L. Jacobson

**Affiliations:** 1Department of Entomology and Plant Pathology, Auburn University, 301 Funchess Hall, Auburn, AL 36849, USA; jwm0055@auburn.edu (J.W.M.); jba0022@auburn.edu (J.B.M.); chr138@msstate.edu (C.R.); ajk0055@auburn.edu (A.K.); 2College of Forestry, Wildlife and Environment, Auburn University, 602 Duncan Drive, Auburn, AL 36849, USA; tds0009@auburn.edu; 3Edisto Research and Education Center, Department of Plant and Environmental Sciences, Clemson University, Blackville, SC 29817, USA; sophiarchristian@gmail.com (S.R.C.); greene4@clemson.edu (J.K.G.); 4Delta Research and Extension Center, Mississippi State University, Stoneville, MS 39762, USA; wdc165@msstate.edu (W.C.); jgore@drec.msstate.edu (J.G.); 5Department of Entomology and Plant Pathology, North Carolina State University, 3210 Ligon St., Raleigh, NC 27695, USA; gkennedy@ncsu.edu; 6Department of Entomology, Texas A&M University, College Station, TX 77843, USA; david.kerns@ag.tamu.edu; 7Virginia Tech, Tidewater Agricultural Research and Extension Center, Suffolk, VA 23437, USA; smalone@vt.edu (S.M.); svtaylor@vt.edu (S.T.); 8West Florida Research and Education Center, Department of Entomology and Nematology, University of Florida, Jay, FL 32565, USA; paula.moraes@ufl.edu; 9Department of Entomology, University of Georgia, 2360 Rainwater Rd., Tifton, GA 31793, USA; proberts@uga.edu (P.R.); mtoews@uga.edu (M.T.); 10Department of Entomology and Plant Pathology, University of Tennessee, Knoxville, TN 37996, USA; sdstewart@utk.edu

**Keywords:** Aphididae, *Aphis gossypii*, Solemoviridae, CLRDV, crop, vector-borne, pathogen, vector ecology

## Abstract

**Simple Summary:**

Cotton leafroll dwarf virus (CLRDV) is capable of causing yield loss in cotton. Eight species of aphids have been reported to feed on cotton, but *Aphis gossypii* is the only known CLRDV vector in the United States (U.S.). Little is known about their distribution, abundance, and seasonal dynamics in the southern U.S. The epidemiological value of understanding this prompted a two-year study to monitor the populations of aphids that infest cotton fields throughout the southern U.S., where CLRDV has been reported. *Aphis gossypii* and *Protaphis middletonii* were the most abundant aphid species collected. *Aphis craccivora*, *Macrosiphum euphorbiae*, *Myzus persicae*, *Rhopalosiphum rufiabdominale*, and *Smynthurodes betae* were also detected in this study; however, their populations remained consistently low throughout the collection period. Results from this study presented novel information regarding the seasonal variation of the species and populations of aphids associated with cotton in the region.

**Abstract:**

Cotton leafroll dwarf virus (CLRDV) is an emerging aphid-borne pathogen infecting cotton, *Gossypium hirsutum* L., in the southern United States (U.S.). The cotton aphid, *Aphis gossypii* Glover, infests cotton annually and is the only known vector to transmit CLRDV to cotton. Seven other species have been reported to feed on, but not often infest, cotton: *Protaphis middletonii* Thomas, *Aphis craccivora* Koch, *Aphis fabae* Scopoli, *Macrosiphum euphorbiae* Thomas, *Myzus persicae* Sulzer, *Rhopalosiphum rufiabdominale* Sasaki, and *Smynthurodes betae* Westwood. These seven have not been studied in cotton, but due to their potential epidemiological importance, an understanding of the intra- and inter-annual variations of these species is needed. In 2020 and 2021, aphids were monitored from North Carolina to Texas using pan traps around cotton fields. All of the species known to infest cotton, excluding *A. fabae*, were detected in this study. *Protaphis middletonii* and *A. gossypii* were the most abundant species identified. The five other species of aphids captured were consistently low throughout the study and, with the exception of *R. rufiabdominale*, were not detected at all locations. The abundance, distribution, and seasonal dynamics of cotton-infesting aphids across the southern U.S. are discussed.

## 1. Introduction

The spatiotemporal distribution and abundance of insect vectors can dramatically influence the spread and prevalence of vector-borne disease in agricultural crops [1,2,3,4,5,6,7,8,9,10]. For a given set of host, pathogen, and environmental conditions, the epidemiology of a vector-transmitted pathogen can be highly dependent on the population dynamics, behavior, mode of transmission, and landscape-level movement of the vector [11]. Vector abundance often varies inter- [1,2,3,4,5,6,7,9,12] and intra-annually [2,7,9,12], which may consequently influence vector-borne disease spread. Because the timing of pathogen inoculation can impact disease spread and the likelihood of infection [13,14], a further understanding of intra-annual variation in vector populations throughout the growing season is needed. This study was conducted to understand the distribution, abundance, and seasonal dynamics of *Aphis gossypii* Glover, the reported vector of cotton leafroll dwarf virus (CLRDV; family Solemoviridae, genus *Polerovirus*) in cotton (*Gossypium hirsutum* L.), and an additional seven species of aphids reported to occasionally infest cotton in the U.S.

CLRDV is the causal agent for cotton blue disease, which was previously observed in Africa, Asia, and South America [15,16,17,18,19,20,21]. CLRDV is the first virus reported to cause yield loss in cotton in the southeastern United States (U.S.), and has been detected in Alabama (AL), Georgia (GA), Florida (FL), Louisiana, Texas (TX), Mississippi (MS), South Carolina (SC), North Carolina (NC), Arkansas, Tennessee (TN), Virginia (VA), Oklahoma, and Kansas [22,23,24,25,26,27,28,29,30,31,32,33,34]. CLRDV incidence has been reported to be highly variable across the southern U.S. based on symptomatology. CLRDV monitoring has been complicated by the use of disease symptoms that overlap with those caused by other agronomic stresses, and has not included asymptomatic infections detected using RT-PCR. In AL and GA, the CLRDV incidence of 60–100% has been reported in cotton using RT-PCR [35] and has also been reported in various weed species [36,37]. However, it is not clear what hosts the vector is coming from prior to colonizing cotton and the role these weeds may serve in the epidemiology of CLRDV. The high incidence suggests that there are large populations of vectors, virus, or both that are contributing to virus spread in these landscapes [38].

Eight species of aphids are reported to infest cotton in the U.S.: cotton aphid, *A. gossypii*; corn root aphid, *Protaphis middletonii* Thomas; cowpea aphid, *Aphis craccivora* Koch; bean aphid, *Aphis fabae* Scopoli; potato aphid, *Macrosiphum euphorbiae* Thomas; green peach aphid, *Myzus persicae* Sulzer; rice root aphid, *Rhopalosiphum rufiabdominale* Sasaki; and bean root aphid, *Smynthurodes betae* Westwood [39]. Of these, only *A. gossypii* is reported to annually infest cotton, cause direct feeding injury, and transmit CLRDV to cotton in a persistent circulative, non-propagative manner [40,41,42]. Viruliferous alatae of *A. gossypii* are able to transmit CLRDV in less than 15 min and for up to 23 days [41,43]. *Aphis gossypii* has primarily been monitored during weeks that it infests cotton [44,45,46,47,48], but now there is a need to understand season-long dynamics to provide insight on the potential for virus spread at other times of the growing season, and to determine which aphid species are abundant in cotton agroecosystems. Preliminary data from a one-year trapping study in AL showed that CLRDV virus spread occurred mid-season, concurrent with *A. gossypii* flights, as well as early May and late August when *A. gossypii* was not abundant [35]. The seven other species are not regularly observed on cotton in the U.S. [39]. Any of these seven species could serve as transient vectors through short-term feeding. Little is known about the seven species’ distribution, abundance, or seasonal dynamics across the U.S. Cotton Belt. All seven species have been reported to transmit at least one plant virus [49]. The vector competence of five of these species to transmit CLRDV is unknown. Two of these species, *M. persicae* and *A. craccivora*, are reported to spread CLRDV to chickpea in India [20,50], but did not successfully transmit CLRDV to cotton in the U.S. [43].

The epidemiological importance of understanding the distribution, abundance, and seasonal dynamics of these eight species warranted a study to monitor their populations during the cotton-growing season in areas where CLRDV has been detected. The objectives of this study were to: (1) characterize the season-long presence, distribution, abundance, and composition of the eight aphid species reported to infest cotton; and (2) quantify and compare each species’ intra- and inter-annual variation among locations. These objectives were assessed by trapping the eight aphid species around cotton fields in the southern U.S. during the 2020 and 2021 growing seasons.

## 2. Materials and Methods

### 2.1. Aphid Monitoring

In 2020 and 2021, dispersing aphid alatae were collected from sites located in cotton agroecosystems in AL, FL, GA, MS, NC, SC, TN, TX, and VA (for a total of 11 locations, Table 1). Alatae were captured weekly using yellow pan traps filled with 50% propylene glycol solution. Each week, aphids were collected from traps and stored in 70% ethanol until identification; see Mahas et al. (2022) for description of traps and methods. Four traps were evenly distributed along field borders of a cotton field (one trap per side of field) for each site except in FL, where 16 traps were used. In NC, traps were placed near field corn and vegetable crops instead of cotton. Aphids were trapped throughout the cotton-growing season until defoliation occurred (Table 1). Initiation dates varied by planting date among locations, and termination dates for trapping varied based on personnel, logistical, and other COVID-19 pandemic-related research restrictions.

Alatae for the eight aphid species were counted and identified [39], see Mahas et al. (2022) for voucher specimens and equipment used. Other species were recorded as ‘other’. If there were 25 or fewer aphids in a given trap, then all specimens were identified. When the number of aphids per trap exceeded 25, a random subsample of 25 aphids were identified to estimate species abundance [12,51]. Aphids stored in ethanol were poured into a coded, gridded Petri dish, and swirled to randomly distribute aphids. Grid codes were put through a list randomizer. The generated list was followed so that a random sample of 25 alatae were collected from the Petri dish. Estimated species abundance was calculated by multiplying the proportion of each species identified in the sub-sample by the total number of alatae collected in the trap sample.

### 2.2. Statistical Analyses

The season-long distribution, abundance, and composition of the cotton-infesting aphid species were characterized via maps and line graphs. Each cotton-infesting aphid species’ total estimated abundance for the traps at each location within a given year was calculated in Excel. This, along with the global positioning system (GPS) coordinates that were collected at each location, were imported into ArcGIS Pro 3.0.3. The Chart Symbology feature was used to display the proportional abundance of each aphid species compared to the total cotton-infesting aphid species collected at each location for 2020 and 2021. To compare each aphid species’ populations throughout the collection period, the SummarySE function in R Studio (version 4.1.2; R Foundation, Vienna, Austria) was used to calculate the means and ±95% confidence intervals (CIs) of the total number of aphids captured in the traps at each location. The ggplot function in R was used to display these data through line graphs.

Statistical analyses were conducted to quantify and compare the effect of location, week, year, and their interaction terms on the abundance of aphids. Datasets from some locations were truncated before analysis, so the same trapping weeks for each year were represented in the dataset. In order to maximize the number of weeks analyzed, MS and FL were excluded from the analysis because they only had one and zero weeks, respectively, in common with the other locations for both years (Table 1). As a result, seven weeks of trapping data for each year from southern AL, GA, SC, TX, northern AL, central AL, NC, VA, and TN were included for analysis. Each model included aphid species’ abundance as the dependent variable, and the independent variables were location, week, year, and all of the two- and three-way interaction terms. Species abundance data were log transformed using log(y + 1) to achieve normality and analyzed using the lm function in R Studio. The anova() function in R was used to conduct an analysis of variance (ANOVA) using type II sum of squares from the results of the lm to assess the effect of each categorical variable and their interaction terms on the dependent variable. Statistical analyses were only conducted on *A. gossypii* and *P. middletonii.* No analyses were conducted for *M. persicae*, *A. craccivora*, *R. rufiabdominale*, and *M. euphorbiae* due to the low counts, and no *S. betae* or *A. fabae* were detected during this timeframe.

## 3. Results

To characterize the season-long distribution, abundance, and composition of each species of cotton-infesting aphids, their populations were monitored weekly throughout the collection period. A total of 63,780 alatae were collected in this study. In 2020, 36,098 alatae were collected and classified as ‘other’ (50.89%), *P. middletonii* (42.35%), *A. gossypii* (4.06%), *R. rufiabdominale* (1.26%), *M. euphorbiae* (0.81%), *A. craccivora* (0.42%), *M. persicae* (0.11%), and *S. betae* (0.10%). In 2021, 27,682 alatae were collected and classified as “other” (56.62%), *P. middletonii* (31.66%), *A. gossypii* (8.55%), *R. rufiabdominale* (1.49%), *M. euphorbiae* (0.67%), *M. persicae* (0.51%), *A. craccivora* (0.47%), and *S. betae* (0.03%). *Aphis fabae* was not detected in this study. *Protaphis middletonii* was the most abundant species of interest present across all locations for both years, except in GA during 2020 and FL during 2021, where *A. gossypii* was the most abundant (Figure 1). *Aphis gossypii* was the second most abundant species each year, while the five remaining species together accounted for less than 4% of the relative species composition each year.

*Aphis gossypii* was detected at all locations and was less abundant toward the beginning of the growing season (Figure 2). Peaks in *A. gossypii* abundance occurred in southern AL, central AL, and GA in late June to mid-July, which is also when and where *A. gossypii* was the most abundant throughout the study each year. Smaller peaks in SC, VA, TX, and FL populations also occurred during this time for one of the two years (Figure 2). *Protaphis middletonii* was more abundant in southern AL, central AL, northern AL, and NC, and was detected at all locations throughout the collection period. Populations of *P*. *middletonii* tended to peak in April and May, except for northern AL, where it peaked in August each year (Figure 3A and Figure 4A). Populations of the five other aphid species remained consistently low throughout the study across all locations and both years. During the collection period, *S. betae* was detected in southern AL, SC, VA, and TN (Figure 3B and Figure 4B); *M. persicae* was detected in southern AL, central AL, northern AL, SC, NC, FL, GA, and VA (Figure 3C and Figure 4C); *A. craccivora* was detected in southern AL, central AL, northern AL, SC, NC, FL, GA, TN, and MS (Figure 3D and Figure 4D); *R. rufiabdominale* was detected at all locations (Figure 3E and Figure 4E); and *M. euphorbiae* was detected in southern AL, central AL, northern AL, SC, NC, GA, TN, TX, and VA (Figure 3F and Figure 4F).

An ANOVA of *A. gossypii* counts, using type II sum of squares, showed significant differences among week, location, and year (Table 2). We also found significant two- and three-way interactions among all the variables (Table 2), indicating that differences among weeks depended on location and year, differences among locations depended on year and week, and differences among years depended on location and week. An ANOVA of *P. middletonii* counts using type II sum of squares showed a significant difference among week and location, but not year (Table 2). There were also significant two- and three-way interactions among all the variables (Table 2), indicating that differences among weeks depended on location and year, differences among locations depended on year and week, and differences among years depended on location and week.

## 4. Discussion

While previous research monitored for the species of aphids known to infest cotton during a single year in southern AL and GA [35], this is the first study to assess their abundance, distribution, and seasonal dynamics across the southern U.S. in a multi-year study. Seven of the eight aphid species reported to feed on cotton were detected in this study. With the addition of *S. betae* captured in southern AL, VA, SC, and TN, these were the same species detected in the study by Mahas et al. (2022). Overall, *P middletonii* and *A. gossypii* were the most abundant species each year. The abundance of the five additional species of aphids was consistently low throughout the study. The finding that over 50% of aphids captured were ‘other’ is not surprising as our trapping methods attract many different species and this study only focused on eight aphid species.

*Aphis gossypii* was captured at all locations throughout most of the growing season. The largest trap captures or “flights” of *A. gossypii* were observed in late June to mid-July in southern AL, central AL, and GA where flights were at least 3-fold greater than the largest observed in VA, NC, SC, TN, northern AL, MS, FL, or TX. This is consistent with findings from previous studies reporting large populations of *A. gossypii* for this geographic area [35,44,45,46,47,48]. Such information is valuable in the context of understanding the epidemiology of CLRDV, because it shows a greater abundance of the vector in areas with a higher reported incidence of CLRDV. A companion study examining CLRDV spread in relation to aphid flights in southern AL, central AL, and northern AL showed that virus spread was detected over a greater number of weeks in southern and central AL than northern AL, where fewer *A. gossypii* were observed [52]. Resources were not available to monitor virus spread at all sampled locations; however, taken together, the results of these studies show that the primary vector of CLRDV has a variable abundance from VA to TX. The interacting vector, host, environment, and pathogen dynamics [11] will likely influence the incidence of CLRDV across this area, and future research is needed to correlate CLRDV incidence with vector abundance.

*Protaphis middletonii* was the most abundant species at all locations each year with the exceptions of GA during 2020 and FL during 2021, where *A. gossypii* was the predominant species present. The abundance of *Protaphis middletonii* may be linked to common crop and weedy hosts (e.g., cotton, corn, crabgrass, pigweed, purslane, ragweed) near sampling locations. This species is often tended by ants whose protection may allow for larger population sizes [53,54]. Populations of *P. middletonii* have been observed to peak earlier in the cotton-growing season than *A. gossypii* [35]. Mahas et al., 2022, also showed that some CLRDV spread coincided with the peak flights of *P. middletonii*, which was also observed in our companion manuscript [52]. These observations provide rationale for testing the vector competence of this species, but it would not help to explain why the high incidence of CLRDV appears to be restricted to southern portions of Alabama and Georgia.

Location, week, year, and all of the two- and three-way interaction terms were statistically significant in the analyses of *A. gossypii* abundance. The same was observed for *P. middletonii* abundance, except year was not significant. The underlying factors contributing to these patterns cannot be discerned from this study and may involve many different components. The observed distribution and abundance of species is inherently a spatiotemporal process driven by multiple parameters [55], including gradients of temperature and precipitation, topographical features (e.g., water bodies, mountains, valleys, urban structures, hills, etc.), and vegetation patterns [56,57]. Weather-related factors impact the reproductive mode of aphids [58], development [59], survival [60], phenology [61], and timing of flights [62], which, in turn, can influence their distribution in space and time [63,64,65]. The spatiotemporal distribution, abundance, and composition of host plants in the landscape can also affect the seasonal dynamics of aphid populations, and consequently, the distributions of pathogens they transmit [1,2,4,5,66,67,68,69,70,71,72,73]. Some of these factors may change temporally, which can lead to intra- and inter-annual variations in community composition [2,69]. Based on the level IV ecoregion scale, each of our trapping sites were located in a different ecoregion, which may be indicative of the wide variation in the distribution, abundance, and seasonal dynamics of aphid populations observed in this study [57]. Factors such as these should be investigated in future studies to better understand what may be driving vector abundance and CLRDV incidence in cotton.

These findings can serve as baseline data regarding the abundance, distribution, and seasonal dynamics of cotton-infesting aphid species in the southern U.S., where little to no prior information is available. *Aphis gossypii* is the only vector known to transmit CLRDV to cotton and the only aphid species to annually infest cotton in the U.S., whereas the other species identified in this study are rarely observed in cotton, especially the root-feeding species. Although potential exists for the other aphid species to serve as transient vectors, five of them were rarely detected in this study, suggesting that their contribution to CLRDV spread to cotton may be minimal. Future research identifying additional vectors of CLRDV, and investigating factors driving aphid abundance, seasonal dynamics, and distribution could contribute to a better understanding of regional variation in CLRDV incidence and may be informative for future CLRDV management practices.

## Figures and Tables

**Figure 1 insects-14-00639-f001:**
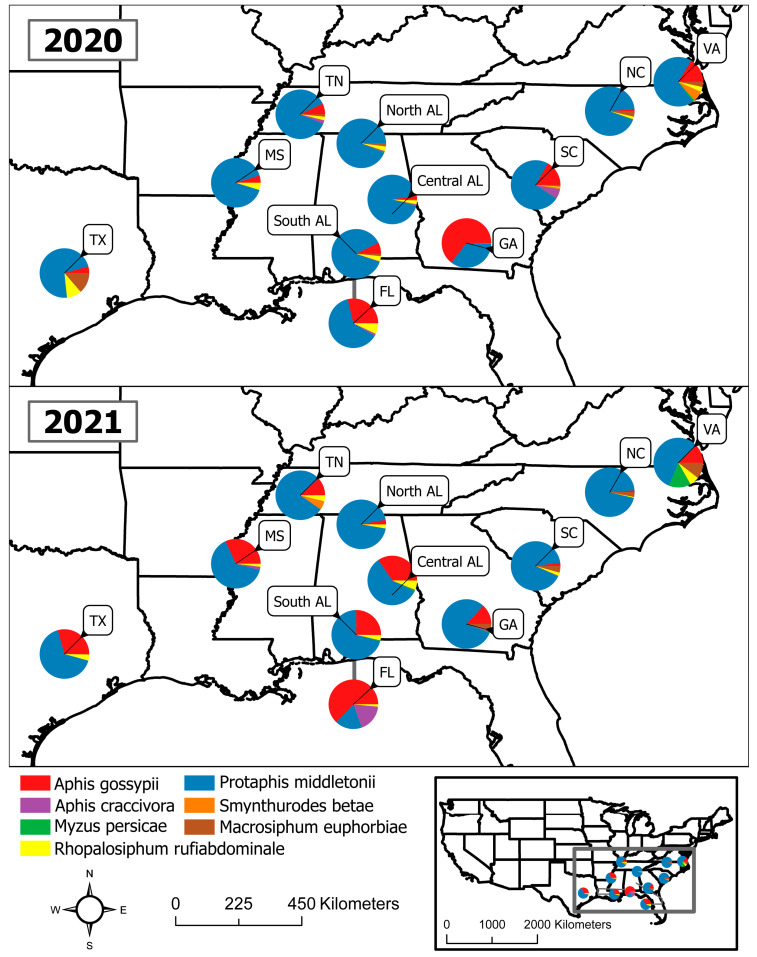
Pie chart shading and segments depict the proportional abundance for each cotton-infesting aphid species captured at each location in 2020 and 2021.

**Figure 2 insects-14-00639-f002:**
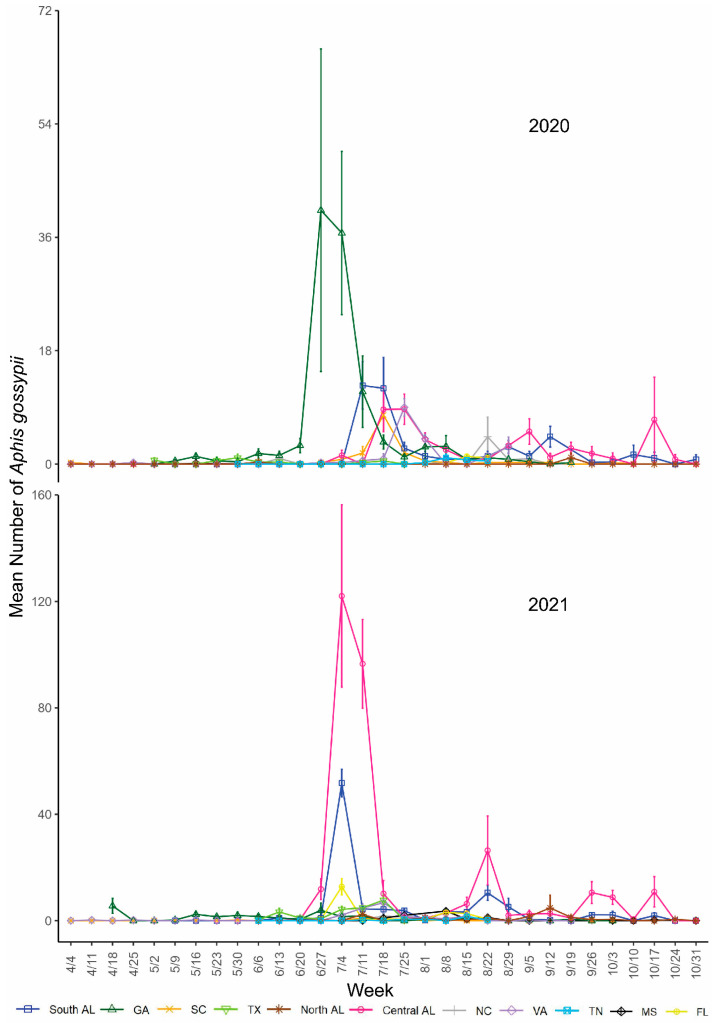
Mean (±95% CI) number of *Aphis gossypii* collected during the 2020 and 2021 cotton-growing seasons. Y-axis ranges differ between graphs.

**Figure 3 insects-14-00639-f003:**
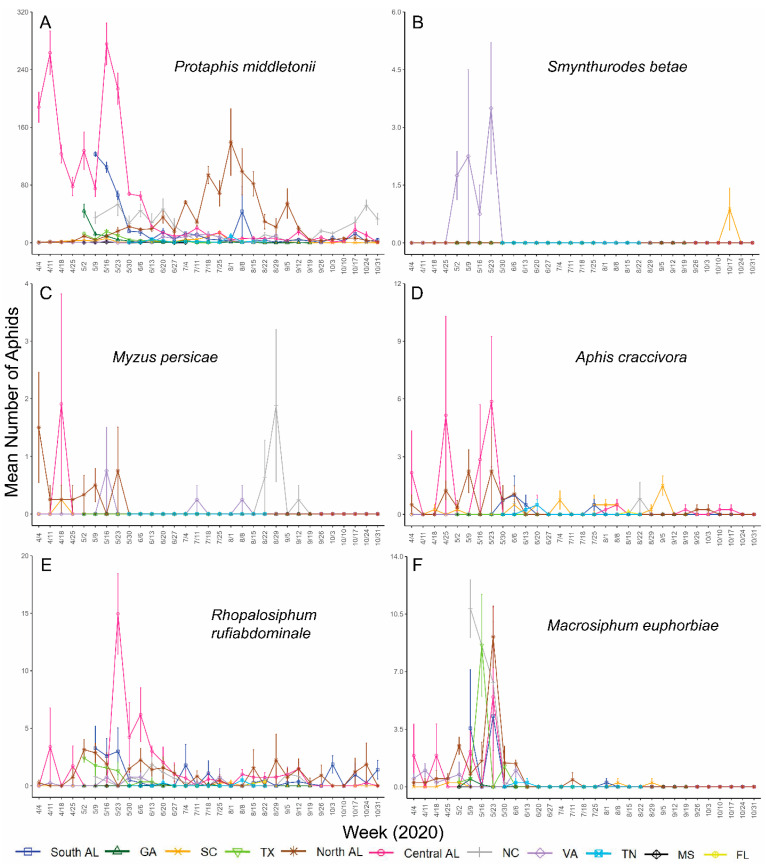
Mean (±95% CI) number of (**A**) *Protaphis middletonii*, (**B**) *Smynthurodes betae*, (**C**) *Myzus persicae*, (**D**) *Aphis craccivora*, (**E**) *Rhopalosiphum rufiabdominale*, and (**F**) *Macrosiphum euphorbiae* collected at each location during the 2020 cotton-growing seasons. Y-axis ranges differ between graphs.

**Figure 4 insects-14-00639-f004:**
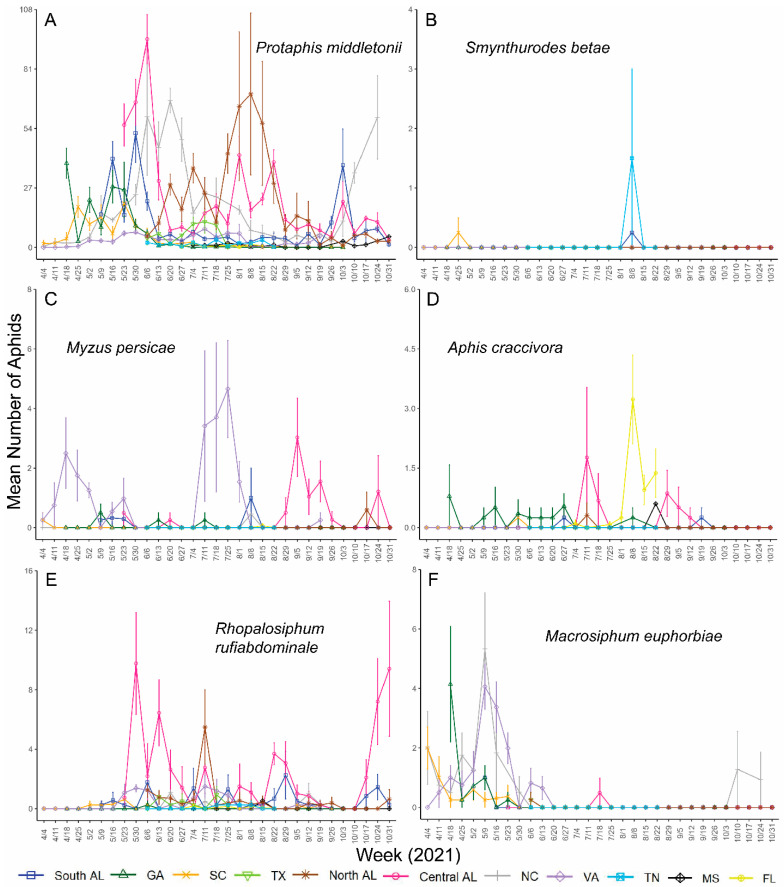
Mean (±95% CI) number of (**A**) *Protaphis middletonii*, (**B**) *Smynthurodes betae*, (**C**) *Myzus persicae*, (**D**) *Aphis craccivora*, (**E**) *Rhopalosiphum rufiabdominale*, and (**F**) *Macrosiphum euphorbiae* collected at each location during the 2021 cotton-growing seasons. Y-axis ranges differ between graphs.

**Table 1 insects-14-00639-t001:** Date ranges when aphids were collected for each location during the 2020 and 2021 cotton-growing seasons.

City, State	Location ^a^	Date Range (2020) ^b^	Date Range (2021) ^b^
Brewton, Alabama	South AL	9 May–31 October	9 May–31 October
Tifton, Georgia	GA	2 May–19 September	18 April–3 October
Blackville, South Carolina	SC	4 April–31 October	4 April–1 October
College Station, Texas	TX	2 May–25 July	6 June–25 July
Belle Mina, Alabama	North AL	4 April–31 October	1 June 31 October
Shorter, Alabama	Central AL	4 April–31 October	23 May–31 October
Apex, North Carolina	NC	9 May–31 October	4 April–24 October
Suffolk, Virginia	VA	4 April–29 August	4 April–19 September
Jackson, Tennessee	TN	30 May–22 August	6 June–22 August
Stoneville, Mississippi	MS	2 May–4 July	4 July–31 August
Jay, Florida	FL	8 August–22 August	27 June–22 August

^a^ Names of the field sites where aphids were collected. ^b^ When pan trap samples were collected.

**Table 2 insects-14-00639-t002:** ANOVA results using type II sum of square errors to assess the effect of week, location, year, and all of the two- and three-way interaction terms on *Aphis gossypii* and *Protaphis middletonii* abundance.

Dependent Variable	Independent Variable	Sum of Squares	df	*F*	*p*
*Aphis gossypii*	Week	74.12	6	42.00	<0.001
	Location	119.64	8	50.86	<0.001
	Year	7.22	1	24.55	<0.001
	Week/Location	97.05	48	6.88	<0.001
	Week/Year	11.69	6	6.63	<0.001
	Location/Year	70.91	8	30.14	<0.001
	Week/Location/Year	97.31	48	6.89	<0.001
*Protaphis middletonii*	Week	18.28	6	7.94	<0.001
	Location	470.30	8	153.26	<0.001
	Year	0.71	1	1.85	0.170
	Week/Location	98.60	48	5.36	<0.001
	Week/Year	6.16	6	2.68	0.015
	Location/Year	43.03	8	14.02	<0.001
	Week/Location/Year	58.55	48	3.18	<0.001

## Data Availability

The data presented in this study are available on request from the corresponding author.

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
