# Peer review of "The Spatiotemporal Distribution, Abundance, and Seasonal Dynamics of Cotton-Infesting Aphids in the Southern U.S."

_insects, 2023, doi:10.3390/insects14070639_

Round 1

Reviewer 1 Report

Although this article is correct in its form, there are many questions that researchers must ask themselves. Carrying out a separate analysis of the flight and abundance of the vectors has many risks, since the viral incidence may vary from year to year depending on many factors that the authors do not take into account. The authors should consider other studies such as weeds as alternative hosts in the absence of cotton crop and as well as knowing where the viral inoculum comes from each year? Where do those aphids that occupy the cotton crop come from? I also think they should do some random sampling on the plants to observe populations of wingless aphids. Currently, studies on the movement of aphids are carried out with landscape studies such as this one:

Clemente-Orta, G., Albajes, R., & Achon, M. A. (2020). Early planting, management of edges and non-crop habitats reduce potyvirus infection in maize. Agronomy for Sustainable Development40, 1-12.

Clemente-Orta, G., Albajes, R., Batuecas, I., & Achon, M. A. (2021). Planting period is the main factor for controlling maize rough dwarf disease. Scientific reports11(1), 1-12.

they could use the reference in the discussion to broaden their future line and justify the need to expand the scales of study. There are many variables that can affect and influence these aphids. The fact that the p-values have a value of <0.001 makes me think that there is overdispersion in the models as well as doubt that these variables really have that effect on abundance.

I am also very surprised that the other category accounts for more than 50% of insects caught and the % of A. gossypii is quite low. maybe they could have used other types of traps like yellow sticky traps?

In conclusion, it is a very basic study but correct in its form.

minnor comments:

Line 39: change Protaphis middletonii to P. middletonii

Line 74: change Aphis gossipy to A. gossypii

Reviewer 2 Report

Here the authors analyzed cotton-infesting aphid abundance in several locations across southern U.S. The main finding was that Aphis gossypii and Protaphis middletonii were the most abundant aphid species collected and that the primary vector, A. gossypii abundance varied across locations and years. While the data the new and informative, I feel the analysis can be strengthened by including virus incidence data in the paper (not the companion paper mentioned by the authors) to make more meaningful connections between vector abundance and disease epidemiology. Alternatively, it would have been useful to develop a prediction model using climatic and landscape variables as predictors of aphid abundance. Since the authors already collected data on other aphid species and it is not clear that other species do not transmit CLRDV, I suggest analyzing the effect of aphid species richness and evenness as well. I have some minor edits in the attached pdf.
